# Early Intervention for Children at High Risk of Developmental Disability in Low- and Middle-Income Countries: A Narrative Review

**DOI:** 10.3390/ijerph16224449

**Published:** 2019-11-13

**Authors:** Maya Kohli-Lynch, Cally J. Tann, Matthew E. Ellis

**Affiliations:** 1Centre for Academic Child Health, University of Bristol, 1-5 Whiteladies Road, Bristol BS8 1NU, UK; M.ellis@bristol.ac.uk; 2Maternal, Adolescent, Reproductive & Child Health, Department of Infectious Diseases Epidemiology, London School of Hygiene & Tropical Medicine, London WC1E 7HT, UK; cally.tann@lshtm.ac.uk; 3Neonatal Medicine, University College London Hospitals NHS Foundation Trust, London NW1 2BU, UK; 4MRC/UVRI & LSHTM Uganda Research Unit, Entebbe P.O.Box 49, Uganda

**Keywords:** developmental disability, low- and middle-income countries, early childhood intervention, newborn

## Abstract

In low- and middle-income countries (LMICs), while neonatal mortality has fallen, the number of children under five with developmental disability remains unchanged. The first thousand days are a critical window for brain development, when interventions are particularly effective. Early Childhood Interventions (ECI) are supported by scientific, human rights, human capital and programmatic rationales. In high-income countries, it is recommended that ECI for high-risk infants start in the neonatal period, and specialised interventions for children with developmental disabilities as early as three months of age; more data is needed on the timing of ECI in LMICs. Emerging evidence supports community-based ECI which focus on peer support, responsive caregiving and preventing secondary morbidities. A combination of individual home visits and community-based groups are likely the best strategy for the delivery of ECI, but more evidence is needed to form strong recommendations, particularly on the dosage of interventions. More data on content, impact and implementation of ECI in LMICs for high-risk infants are urgently needed. The development of ECI for high-risk groups will build on universal early child development best practice but will likely require tailoring to local contexts.

## 1. The Challenge 

Around the world, almost half of all deaths in children under five occur in the newborn period [1,2]. Ninety-nine percent of newborn deaths are in low- and middle-income countries (LMICs) and prematurity, intrapartum-related neonatal deaths (‘birth asphyxia’), sepsis and meningitis account for 75% of these [2]. As improved obstetric and neonatal care is scaled up in LMICs, the number of children surviving these neonatal conditions at high risk of neurodevelopmental impairment and disability increases with an estimated risk of around 40% of at least one impairment in any developmental domain after perinatal insults. Intrapartum-related neonatal encephalopathy alone causes an estimated 42 million disability-adjusted life years [3,4,5].

A systematic analysis for the Global Burden of Diseases, Injuries, and Risk Factors Study 2016 estimated that there are 52.9 million children under five with developmental disabilities, 95% of whom live in LMICs, with little evidence of change since the 1990s [6]. This is likely to be an underestimate due to under-reporting of cerebral palsy (a major cause of childhood disability worldwide) and Attention Deficit Hyperactivity Disorder (ADHD), and the lack of weighting adjustment of Years Lived with Disability for children living in LMICs where few services are available and stigma often more explicit [7,8].

With the UN Sustainable Development Goals (2015–2030) and the Global Strategy for Maternal, Adolescent and Child Health advocating the need to address newborns thriving beyond survival, global child health priorities now include improving the lives of survivors of conditions such as prematurity, neonatal encephalopathy and neonatal infections [9]. Intervention is particularly beneficial during the first thousand days of life due to rapid brain growth and neuroplasticity, with impact demonstrated on child development as well as educational achievements and higher earnings later in life [10,11,12]. The WHO, UNICEF and the World Bank’s Nurturing Care Framework recognise three ‘levels’ of interventional support for child development in the pre-school period: (1) ‘universal’, delivered by society more broadly, e.g., through policy, (2) ‘targeted’, support for children at risk of developmental delay or disability, e.g., community health worker home visits to young mothers, and (3) ‘indicated’, specialised services for children with specific additional needs e.g. community-based groups for caregivers of children with disabilities, previously referred to as CBR (see below) [13]. 

The International Classification of Functioning, Disability and Health (ICF) provides a useful framework for understanding disability; structural and functional impairments, and resultant activity and participation limitations influenced by personal and environmental factors [14]. The ICF framework and WHO Community-Based Rehabilitation (CBR) guidelines are valuable reference tools in the development of early childhood interventions (ECI) and their use is advocated by Yousafzai et al. in their review of interventions for children with disabilities in LMICs [15]. Nonetheless, other approaches can and have been used to successfully develop ECI.

CBR arose from the primary care movement advocated at Alma Ata and has been described as ‘a strategy within general community development for rehabilitation, equalization of opportunities, and social inclusion of all people with disabilities...implemented through the combined efforts of people with disabilities themselves, their families and communities, and the appropriate health, education, vocational, and social services’ [16]. Given this strongly participatory philosophy, it describes a wide range of community initiatives which are situation-specific. Evaluating CBR has proved challenging and the approach is only supported by small numbers of low-quality studies [17,18]. Scanty scientific literature on CBR is thought to be largely due to the variability of initiatives and lack of planning and resources, both financial and human, committed to evaluation from the outset [19,20]. In the absence of robust and compelling evidence of impact, it has struggled to achieve sustainable funding [17]. Ten years ago, an important review of the whole field of prevention and management of childhood disabilities and impairments in LMICs concluded that more focused higher quality research was needed for progress to be made [21].

The Nurturing Care Framework provides just such an opportunity in which to implement programmatic research for this most marginalised of child populations. The impacts of ‘indicated’ interventions, known as ECI, for infant survivors of perinatal adversity at high risk of developmental disability are less researched and receive significantly less donor funding compared to ‘universal’ and ‘targeted’ early child development (ECD) interventions [22]. The evidence base for ECI for high-risk children in LMICs, a population often triply disadvantaged by disability, poverty, and stigma, is now being co-created by researchers and communities and will be explored in this review [13,22,23].

The aim of this narrative review is to provide a broad overview of the current evidence on ECI in LMICs [24]. We searched PubMed, Google Scholar and the ISCRTN Trial Registry for studies on early intervention for children at high risk of developmental disability in LMICs and used snowball referencing to identify further relevant studies. The literature search generated five relevant studies, discussed in Section 2.4. 

## 2. Current Evidence on Early Intervention for Children at High Risk of Developmental Disabilities

### 2.1. Why—The Rationale for Investment in Early Childhood Intervention

There are several rationales for investment in ECI for children with developmental disabilities and their families [25]. Firstly, as stated above, the first thousand days are evidence-based to be critical for neurodevelopment; therefore, intervening at the time of greatest neuroplasticity is likely to have the biggest impact (scientific rationale) [10]. Secondly, the human rights rationale: every child has the right to achieve their full developmental potential and must be supported in this by their caregiver, family, community and wider society, as stated in the ‘UN Convention on the Rights of the Child’. Thirdly, gains in human capital and a resultant more productive, less impoverished, workforce has been used to convince policymakers to invest in ECD more generally (‘human capital’ rationale) [26]. However, such ‘universal’ ECD programmes frequently exclude children with developmental disabilities, even though provision of an enabling environment is undoubtably key in supporting those with developmental disability to meet both their developmental and economic potential. Beyond ECD programming, society as a whole must promote enabling work environments to support individuals in fulfilling their economic potential. In an inclusive and diverse society, inclusion of children with developmental disabilities in both ‘universal’ ECD programming and ‘indicated’ ECI is clearly warranted. Finally, ECI for young children with disabilities can improve child wellbeing, allow parents more time to engage in productive work, reduce the risk of abuse and neglect, and help achieve school readiness (programmatic rationale) [27]. 

### 2.2. When to Implement and How to Reach the Right Children

Interventions during the first thousand days of life can be understood within a ‘primary, secondary and tertiary prevention’ framework. Primary preventative interventions prevent the cause of impairment occurring, e.g., addressing the “three delays” in receiving healthcare in obstetric emergencies to reduce the incidence of intrapartum-related neonatal events [28]. Secondary preventative interventions enhance development, e.g., promotion of parent-infant responsive caregiving [29]. Tertiary preventative interventions minimise comorbidities and factors which influence disability, e.g., anticonvulsants to prevent seizures in children with cerebral palsy. The ECIs described in this review encompass the first thousand days of life and include secondary and tertiary interventions; primary interventions fall outside the remit of this review. 

There is some evidence that developmental care and Kangaroo Mother Care, both secondary prevention strategies which are implemented as early as the day of birth for preterm newborns, improve health, nutrition and family outcomes [29,30,31]. To implement tertiary prevention interventions for infants with developmental disabilities, early detection of (i) high-risk newborns and (ii) infants with early developmental disabilities is needed. 

Studies variably defined ‘high-risk newborns’ as those requiring resuscitation at birth, those with neonatal encephalopathy, prematurity, low birth weight, jaundice, hypoglycaemia, neonatal meningitis/sepsis, neonatal seizures and congenital anomalies. Neonatal conditions, such as neonatal encephalopathy, prematurity and others have been found to pose substantial neurodevelopmental risks to infants, but there is no consensus in the literature on which criteria to use for entry into ECI, e.g., some studies recruited premature newborns with a gestational age < 37 weeks, while others recruited < 34-week gestation newborns [32,33]. 

A recent review of early diagnosis and management of cerebral palsy by Novak et al. found that in high-income country (HIC) settings, accurate diagnosis can be made from three months of age with Magnetic Resonance Imaging (MRI) plus Prechtl General Movements assessment (video observation of an infant’s movements when awake, calm and lying on their back), or Hammersmith Infant Neurological Examination (HINE), and after five months with MRI, HINE and the Developmental Assessment of Young Children [34]. In LMICs, MRI and Prechtl General Movements are less available, and Novak et al. recommended the use of HINE alone, although data were lacking from LMIC settings. It is recommended that high-risk infants are also screened with a child development assessment screening tool; however, the predictive validity of such tools for infants in LMICs is not yet clear [35,36]. Evidence supports comprehensive child development assessments for all infants screening ‘positive’ for developmental delay, but in practice, this is difficult in LMICs given the in-depth training needed to certify trainers in most comprehensive assessments, as well as the prohibitive cost of assessment kits [36,37]. Furthermore, lack of standardised norms in LMIC settings means that even when comprehensive assessments are performed, results are difficult to interpret. The Global Scale for Early Development Team, led by the WHO, aims to produce internationally standardised and validated ECD assessment tools targeted at both population and programmatic levels; these tools are currently being field-tested in various LMICs. [38]

In summary, it is recommended that ‘targeted’ interventions for newborns at risk of developmental delay begin in the newborn period and ‘indicated’ interventions for newborns with specific additional needs can begin on detection as early as three months of age [13,34]. It is likely that ‘the earlier the better’ regarding timing of intervention, but more data is needed on methods for identification of high-risk newborns and infants with developmental disabilities, and appropriate timing for ECI in LMICs.

### 2.3. What—Defining Early Intervention for Developmental Disability

While definitions vary throughout the literature, a broad definition of early intervention is given in the Handbook of Early Childhood Intervention as “multidisciplinary services provided to children from birth to 5 years of age to promote child health and well-being, enhance emerging competencies, minimise developmental delays, remediate existing or emerging disabilities, prevent functional deterioration and promote adaptive parenting and overall family function” [39]. While curricula content of these interventions has been difficult to evaluate due to poor description in the literature, heterogeneity where described, and lack of standardised quality measures [40], there is increasing evidence to support ‘targeted’ ECD interventions for children at risk of developmental difficulty due to undernutrition or social disadvantage [29]. There is a lack of strong evidence on ‘indicated’ ECI for infant survivors of adverse neonatal conditions, particularly in LMICs. 

Novak et al.’s HIC-focused review recommended that ECI: (1) optimise motor, cognitive and communication outcomes, involving physiotherapy, occupational and speech and language therapy interventions, (2) prevent secondary impairments with medical interventions, and (3) promote caregiver coping and mental health support strategies. ECI studies from HICs largely focusing on preterm and/or low birth weight infants, have found improved cognitive development with high-dosage interventions which combine home visits, child development centres and parent support groups. These ECI comprise didactic teaching on child health, growth and development, and interactive sessions demonstrating games and activities for child development, with a focus on parent–infant interaction [41,42]. In LMICs and particularly in rural areas where medical and allied health professional services are less available, innovative ways of achieving the three objectives stated above and reaching the most vulnerable children must be sought. Furthermore, lessons from CBR suggest interventions be as focused as possible while remaining inclusive to facilitate standardisation and programme evaluation. [19]

### 2.4. What—Emerging Evidence on ‘Indicated’ Early Childhood Interventions in Low- and Middle-Income Countries

Small studies on ECI for children at high risk of developmental disability in LMICs have found promising results, although target population, intervention content, and outcome measures vary considerably. 

In China, Bao et al. conducted a randomized controlled trial of a home visiting programme for full term infants with an Apgar score ≤ 6 at 5 min after birth. The programme comprised peers educating parents on positive stimulation activities. Follow-up assessments found improved cognitive development with a mean MDI 14.6 points higher in the intervention compared to control group (105 vs. 91, *p* < 0.001) at age 18 months [43]. Wallander et al. conducted a randomized controlled trial of a home-based ECI for infants who required resuscitation at birth in India, Pakistan and Zambia; the ‘Partners for Learning’ programme, comprising play and activities for the child, taught to parents during home visits [44,45]. It is noted that infants recruited to Bao and Wallander’s studies may not necessarily have been at ‘high-risk’ of developmental disability as evidence has found that children who required resuscitation at birth do not have any significant difference in development in the first three years of life compared to those born without any need for resuscitation. [46] In an observational study in India, De Souza et al. recruited young children at high risk of developmental disability to a weekly facility-based ECI in which parents were educated on child health and growth monitoring, and positive stimulation for young children, with moderate attendance by caregivers [33]. Benfer et al. are currently conducting a randomized controlled trial of ‘LEAP-CP’ for high-risk newborns in West Bengal. LEAP-CP is a modular peer-to-peer home visiting intervention and includes education on infant health and nutrition, parental mental health, the caregiving environment, and responsive caregiving, including ‘finding joy in your baby’, plus daily activities delivered to the child by the parent [32]. 

The ABAaNA Early Intervention Programme for infants at high risk of developmental disability, adapted from ‘Getting to Know Cerebral Palsy’ for parents of children over two years of age, was developed for infant survivors of neonatal encephalopathy in Uganda and is currently undergoing feasibility and acceptability testing through a pilot feasibility randomised controlled trial [47,48,49,50]. This ten-modular group, participatory, community-based intervention is delivered through community groups of 6–8 caregivers facilitated by ‘expert mothers’ of older children with neurodisability, supplemented by one or two home visits. Modules were developed in partnership with allied health professionals such as physiotherapists and speech and language therapists. Caregiver groups meet for 2–3 hours every 2–4 weeks working together through the facilitated modules [51]. 

‘Juntos’, based on Getting to Know Cerebral Palsy and ABAaNA modules, is an ECI which was developed for children with congenital zika syndrome and their families in Latin America and is currently undergoing pilot testing in Brazil. [52] Juntos, ABAaNA and Getting to Know Cerebral Palsy collaborate through the ‘Ubuntu Hub’ based at the London School of Hygiene & Tropical Medicine (https://www.ubuntu-hub.org/). 

Common themes for intervention content which have emerged from the LMIC ECI literature include (1) peer-to-peer training, (2) promoting responsive caregiving, and (3) preventing secondary complications such as malnutrition. All the studies mentioned above are small with less than 300 child participants but serve as strong foundations for future larger ECI studies. 

### 2.5. Where to Conduct Early Childhood Intervention Programmes?

Since many infants at high risk of developmental disability are identified at birth, neonatal interventions for optimal development, such as Kangaroo Mother Care, as mentioned above, may be initiated on the neonatal unit and continued at home after discharge into the community. For infants with developmental disability identified after the newborn period, emerging evidence suggests that a combination of community-based group sessions and home visits is the most successful delivery setting for ECI [29]. Home visitation programmes require intensive human resources for implementation and may not be feasible in the scale-up phase of ECI [53]. However, it must be considered that children with disabilities and their caregivers may be less able to travel to health or community facilities to receive interventions and focus must be kept on inclusion of the most marginalised children. 

### 2.6. How—Implementing Early Childhood Intervention Programmes

Early child development is an intersectoral field and as such, collaboration between sectors is vital to the delivery of interventions of high quality and coverage [54]. Given that the health sector is likely to have the most contact with young infants, it is most commonly adopted to deliver and coordinate early childhood interventions, beginning with strong referral systems from newborn health facilities and community health workers [54]. More data is needed on the duration and dosage of ECI required for maximal impact [55]. Questions remain regarding the optimal frequency of contact, intensity, and the expense of such interventions. More evidence is needed on implementation at scale in LMICs to understand how to deliver and successfully scale up programmes aimed at this population of children. 

### 2.7. What Next?

A growing evidence base and support from international agencies have provided a strong investment case for ECD programmes’ ability to unlock the developmental potential of millions of children globally. The impact of ECI on children at risk of developmental disabilities is still emerging but has been highlighted repeatedly as an area which requires more attention, including integration into the wider ECD field [56]. The ECIs discussed here seek to avoid the pitfalls of CBR by being co-created with the community in which they are delivered. As a result, each intervention is individual and tailored to the needs of the community it was developed to serve. This is a strength but does pose its own challenges in terms of reproducibility, adaptation, standardisation and evaluation.

There is no clear consensus on which outcome measures to use in ECI studies and many tools lack standardisation and validation in LMICs. A systematic review of contents of outcome measures in childhood cerebral palsy categorised 283 outcome measures according to the International Classification of Functioning, Disability and Health [57]. A similar review of outcomes measures for ECI in LMICs would be a valuable contribution to the field. School enrolment and attendance, for example, a simple measure which is not always achieved in practice, may be considered. More studies are needed to investigate the optimal timing, content and delivery of ECI in LMICs, including human resources, dosage, and location.

### 2.8. Strengths and Limitations

To our knowledge, this is the first review of early childhood intervention for young children at high risk of developmental disabilities in LMICs. As this was not a systematic review, some relevant studies may have been missed.

## 3. Conclusions

There is unprecedented global attention on early child development. ECD must include survivors of perinatal adversity who are at very high risk of developmental disability. In LMICs, there is preliminary evidence that ECI delivered by peers aimed at preventing secondary complications and promoting parental support and child inclusion in society are successful, but outcomes such as nutrition, quality of life, functioning and participation are not universally reported and there is no consensus on which outcomes should be measured. The use of a standardised framework based on the International Classification of Functioning, Disability and Health and CBR, for reporting intervention content and outcome measures would help clarify messages on ECI for infants at high-risk of developmental disability for policymakers and programmers. While an evidence base is growing for ECI in LMICs, larger and longer-term studies must be undertaken to understand fully what, where, when and how to implement to achieve maximal impact in improving quality of life and participation for infants with developmental disabilities and their families. Future systematic reviews on these areas would be a valuable contribution to the field.

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
