# Peer review of "Early Intervention for Children at High Risk of Developmental Disability in Low- and Middle-Income Countries: A Narrative Review"

_ijerph, 2019, doi:10.3390/ijerph16224449_

Round 1

Reviewer 1 Report

The title of the research is concise and indicates the content of the research.

Abstract presents the key features of the review.

Before going to the challenge section, the authors should provide some contextual background about the project. This is a review and I am not clear why the authors didn't show any search strategy to select the papers they reviewed here. Some important research papers on this topic area may be missing while the review was undertaken. I would recommend that the authors should mention in the limitation sections whether they missed any important research papers on this topic. 

Author Response

Many thanks for this helpful review. Please see attachment for our responses. 

Reviewer 2 Report

The focus of this manuscript is on early intervention for children at risk of developmental disability in low- and middle-income countries. The intent is to provide a review of the current literature and make recommendations for future research to address any gaps. Unfortunately there are some significant issues that need to be addressed before the manuscript should be considered for publication.

The authors have not conducted a systematic or even scoping review. They do not provide enough details on search terms, inclusion/exclusion criteria, or number of artifcles included/exclude. In section 2.4, they simply state: "We searched Google Scholar, PubMed and ISCRTN Trial Registry for studies on ECI for children at high risk of DD in LMICs. We identified five studies with details on ECI in LMICs." This is not a sufficient level of detail. Please refer to PRISMA web site for more information (http://prisma-statement.org/). Even if this was not a systematic review (e.g., more of a scoping review), many journals now require a PRISMA figure/flow chart. This may be another useful resource: https://bmcmedresmethodol.biomedcentral.com/articles/10.1186/s12874-018-0611-x.  Although the section categories are appropriate (e.g., 2.1: Why - The rationale for investing in early childhood intervention, 2.2 When to implement and how to reach the right children, etc.), I found that the text in each section did not always support the goal/intent as outlined. For example, section 2.2 began with a repeat of information about why the first 1000 days are critical (seems best in the 'why' section), then goes on to talk about different types of interventions (seems best in the 'what' section). There is pertinent information about when (e.g., newborns versus later), but there is no comparison of studies that looked at different times of intervening. Perhaps this is because no such studies have been conducted in LMICs, but it's hard to tell because there are no details about the search results.  There in no true synthesis of the literature in any of the sections. For example, in section 2.4, there is many a re-stating of the results of each study. The last paragraph begins to hint at "common themes" but does not go into any details about the them. There are no clear recommendations about next steps beyond "more research is needed." There is again a hint that more evaluation data are needed as well as a review of outcome measures, but there is no organized set directives to guide next steps. 

Minor points: Please review text for grammatical errors and readability (e.g., "In China, Bao et al. trialled a home visiting programme for full term infants with an Apgar score ≤ 6 at 5 minutes after birth peers delivered education sessions to parents on positive stimulation activities, follow up assessments showed improved cognitive development with a mean MDI 14.6 points higher in intervention compared to control group (105 vs 91, p<0.001) at age 18 months."). Also, please spell out HICs (high-income countries) the first time it is used in the text. 

Author Response

(The authors gave the same response as above.)

Reviewer 3 Report

This is a very important topic, and the review is competent and accurate.  It doesn't feel cutting edge; but that reaction may simply reflect the fact that I work and breathe this topic all the time.  There is great value in getting this information to readers who work in public health and environmental issues.

My main comment is that the paper has too many acronyms.  Readers should not be expected to master all of the authors' abbreviations.

The paper draws on important conceptual frameworks (universal, targeted, indicated and primary, secondary, tertiary prevention).

The authors use the word "show"; but I would encourage them to use "find" instead, because the purpose of research is to find, not show.

On line 139, the authors state that detection can begin "as early as three months of age"; but this is too late (and is not aligned with statements elsewhere in the paper).

On line 175, the authors discuss a control group.  Was this an experimental study, with a control group?  The methodology underlying the several studies that are described should be briefly clarified.

The authors should be familiar with the Global Scales for Early Development (GSED), an outcome measure for young children ages 0-3, and with Child Indicators Research, a journal that addresses the measurement of child outcomes, along with the Handbook of Child Well-Being.  

Author Response

(The authors gave the same response as above.)

Round 2

Reviewer 2 Report

Thanks to the authors for their attention to earlier comments and recommendations. I believe they had adequately addressed each and feel it the manuscript is now suitable for publication.